# Multicentric Reticulohistiocytosis Associated with an Early Form of Systemic Lupus Erythematosus: A Case Report of a Rare Disease, with Mini Review of the Literature

**DOI:** 10.3390/jcm11216529

**Published:** 2022-11-03

**Authors:** Elena Biancamaria Mariotti, Alberto Corrà, Elisa Lemmi, Lucrezia Laschi, Cristina Aimo, Lavinia Quintarelli, Walter Volpi, Francesca Nacci, Alice Verdelli, Valentina Ruffo di Calabria, Serena Guiducci, Marzia Caproni

**Affiliations:** 1Section of Dermatology, Department of Health Sciences, University of Florence, 50125 Florence, Italy; 2Section of Anatomical Pathology, Department of Health Sciences, University of Florence, 50125 Florence, Italy; 3Rare Disease Skin Unit, Section of Dermatology, Azienda USL Toscana Centro, University of Florence, 50125 Florence, Italy; 4Section of Dermatology, Azienda USL Toscana Centro, 50125 Florence, Italy; 5Section of Rheumatology, Department of Clinical and Experimental Medicine, University of Florence, 50125 Florence, Italy

**Keywords:** multicentric reticulohistiocytosis, autoimmune disease, case report

## Abstract

Multicentric reticulohistiocytosis (MRH) is the most frequently described form of reticulohistiocytosis (RH), and it is classified as a class IIb non-Langerhans cell histiocytosis. It has been designated as multicentric, being characterized by multisystemic involvement. In fact, although mainly involving the skin, along with the joints, it is a systemic inflammatory condition potentially involving every internal organ. As MRH-related skin findings can mimic rheumatoid nodules or Gottron papules, the histopathology of the cutaneous lesions is often necessary for the correct diagnosis. Approximately one-third of MRH patients have confirmed concomitant autoimmune disorders. A wide variety of autoimmune disorders associated with the disease have been reported in the literature, suggesting immune dysfunction as a factor in the pathophysiology of MRH. A case of MRH associated with autoimmune manifestation is reported in the context of a mini review of the literature, with a focus on clinical presentation, treatments, and treatment outcomes. Moreover, eight cases of MRH associated with autoimmune diseases are briefly discussed.

## 1. Introduction

Reticulohistiocytosis (RH) is a group of rare and clinically heterogeneous proliferative disorders of the mononuclear phagocyte system [1,2], including solitary reticulohistiocytoma, generalized reticulohistiocytosis, and multicentric reticulohistiocytosis. Although it has been stated that the diagnosis of RH relies on the analysis of biopsy specimens [3,4], the findings collected by the clinicians during follow-up are equally important in defining each RH clinical sub-type. Thus, as there are no specific laboratory test findings for Multicentric reticulohistiocytosis (MRH), the diagnosis is made upon clinical presentation, skin and/or synovial biopsy, and supportive radiographic findings. Typical histopathological findings include a lymphohistiocytic infiltrate with multinucleated histiocytes and multi-nucleated giant cells, with a ground-glass appearing eosinophilic cytoplasm [5].

Multicentric reticulohistiocytosis is the most frequently described form of RH, and it is classified as a class IIb non-Langerhans cell histiocytosis [5]. It has been designated as multicentric due to its multisystemic involvement. In fact, it is an inflammatory condition invariably involving the skin along with the joints^1^. It can produce skin changes (usually papulonodular eruptions) and mucosal lesions, and it can mimic other rheumatic conditions and cause destructive arthritis. It can also affect the internal organs, such as the lungs (resulting in pleural effusion) [6], the pericardium [7], and the heart (case reports of pericardial effusion and congestive heart failure) [8], in addition to causing rare cases of mesenteric lymphadenopathy and urogenital lesions [9]. It predominantly affects adult Caucasian females in their fifth and sixth decade of life [5,10].

## 2. Case Report

A 50-year-old Caucasian woman was referred to our dermatology clinic for a cutaneous eruption appearing during summer. She presented with an erythematosus, intensely pruritic, papular rash that involved the face, upper body, and arms, sparing the skin underlying the bra laces, in a photo-distributed pattern (Figure 1).

Along with pruritus, she presented with concerns for arthralgia and morning stiffness, for which she was being treated with anti-inflammatory medications. Upon musculoskeletal system examination, our patient exhibited the ulnar deviation of both hands and angular deformity of the metacarpophalangeal (MCF) and interphalangeal (IF) joints. Due to these manifestations, she received a diagnosis of seropositive rheumatoid arthritis (RA) in 2009. In fact, at that time, high titers of inflammatory markers, rheumatoid factor, and anti-cyclic citrullinated peptide (CCP) autoantibodies were also detected. Moreover, active synovitis of the joints of the hands was appreciable according to ultrasound examination. In 2018, she also displayed bilateral skin-colored nodules overlying the MCF and IF joints (Figure 2), which were assessed as rheumatoid nodules. The skin manifestations of the upper body resembled those of lichen amyloidosis, a rare chronic form of cutaneous amyloidosis clinically characterized by the development of pruritic, often pigmented, hyperkeratotic papules on the trunk and the extremities. The nodules were similar to those of erythema elevatum diutinum, a rare chronic dermatosis typically characterized by red–brown or violaceous papules or nodules symmetrically distributed on acral and peri-articular sites. In order to evaluate these diagnostic hypotheses, biopsies of the deltoid papules and periarticular nodules were performed.

The histopathological exam showed histiocytes and multinucleated giant cells in the papillary and reticular dermis, infiltrating between the collagen bundles. The cells were large and had an eosinophilic, finely granular, ground-glass cytoplasm. A few scattered lymphocytes were also present. The overlying epidermis was thinned, with partial loss of the rete ridges. Immunohistochemical investigations showed positivity for CD68 and CD45 and negativity for CD1a and S100 (Figure 3).

These findings, along with the clinical manifestations, were suggestive of MRH. Lichen amyloidosis and erythema elevatum diutinum were excluded, being characterized by the deposition of amyloid in the papillary dermis in the former and by leukocytoclastic vasculitis in the latter.

Her clinical history included: synovitis of the wrists, knees, ankles and metatarsophalangeal joints; Raynaud phenomenon; recurrent aphthous stomatitis, and xerophthalmia, as well as facial erythema previously diagnosed as acne rosacea (Figure 4).

The immunological findings included hypocomplementemia (low C3 and C4), positive rheumatoid factor, and anti-cyclic citrullinated peptides (CCP) antibodies, along with high titers of antinuclear antibodies (ANA) (1:2560), anti-Sjogren’s-syndrome related antigen A (SSA/Ro52/Ro60) antibodies, and anti- centromere protein B (CENP-B) antibodies. The leucocyte formula was characterized by lymphopenia. Hyper-gammaglobulinemia was detected after serum protein electrophoresis. The patient was tested for a variety of other autoantibodies, including anti-dsDNA, anti-Smith (Sm) antibodies, and anti-U1-ribonucleoprotein (U1RNP) antibodies, which were negative. A complete blood count and urinalysis showed no other specific findings.

Other immunoassays, including a Coombs test, a lupus anticoagulant blood test, and myositis blotting, were requested and found negative. Since MRH may also indicate paraneoplastic disease in 15 to 30% of patients, a screening for malignancies that included a complete abdominal ultrasound and thorax computer tomography was conducted, and the results were negative. Our patient also exhibited clinical and laboratoristic manifestations consistent with an underlying autoimmune disorder. In particular, photosensitivity, hypocomplementemia, and leukopenia suggested an early form of systemic lupus erythematosus. Since the evolution to a defined systemic lupus erythematous may occur, we decided to pursue a close follow-up. The patient was treated with non-steroidal anti-inflammatory drugs, associated with courses of corticosteroids at low-medium dosages in the acute phases. When the disease activity worsened, methotrexate (MTX) was added. It was administered subcutaneously 7.5 mg/week then augmented to 15 mg/week. Afterwards, adalimumab 40 mg administered subcutaneously on alternate weeks was added to the MTX. Since after 10 weeks, no benefit was obtained, the dosage was augmented to 80 mg on alternate weeks for 14 weeks. This treatment was interrupted for lack of improvement. In 2015, the disease activity started worsening again. However, the patient had to move to another country, interrupting all treatments. In January 2019, she returned. A course of corticosteroids associated with MTX 7.5 mg/week was administered. However, MTX had to be suspended after a few weeks for gastro-intestinal intolerance (nausea, vomiting, and abdominal pain). In 2020, MTX was replaced by baricitinib 4 mg/die for 9 months. Then, being ineffective, sarilumab 200 mg, administered on alternate weeks, was started. As the introduction of sarilumab corresponded to the appearance of a cutaneous eruption, later known to be a manifestation of MRH, which at the time, could not be excluded as a possible side effect of the drug, sarilumab was suspended.

After a diagnosis of MRH was made, from December 2021 she was started with subcutaneous tocilizumab 162 mg/week, with improvement of both dermatological and articular manifestations (Figure 5). Hydroxychloroquine 200 mg/die was added, as a preliminary diagnosis of early lupus was made.

During follow-up, the Modified Disease Activity Score (DAS28), with three variables, was used. The disease had been increasing over the years, only reaching a stable decreasing phase after treatment with tocilizumab (Figure 6).

## 3. Discussion

MRH is often associated with neoplastic and autoimmune disorders [11,12]. It has been reported that 15–30% of the patients presenting with MRH exhibited evidence of malignancy [5] and that up to one-third of patients develop malignancy synchronously or metachronously with MRH [10]. For this reason, MRH has been hypothesized to be a paraneoplastic condition. It has been outlined thus, that when the diagnosis is made, a screening for unknown cancer is necessary [5]. Associated malignancies are reported to include breast cancer, ovarian adenocarcinoma, ovarian neuroectodermal tumor, squamous cell carcinoma of the lung, cutaneous squamous cell carcinoma, melanoma [5,9], papillary serous endometrial cancer [5,9], nasopharyngeal cancer, and hepatocellular carcinoma. Clinical cases stating an association with cervical carcinoma, colon carcinoma, and pancreatic adenocarcinoma [5] have also been reported [9]. The associated neoplasm is typically of the epithelial origin [13,14,15,16]. Interestingly, it has been reported that in some cases, remission from cancer leads to concurrent remission of MRH [17]. However, a high prevalence of autoimmune diseases are also known to be associated with MRH [5], as it has been reported that approximately one-third of the patients had confirmed concomitant autoimmune disorders [5]. While a single associated rheumatic disease did not stand out, a wide variety of autoimmune disorders were found, suggesting immune dysfunction as a factor in the pathophysiology of MRH^5^. Moreover, it has been proposed that MRH and autoimmune diseases may display a certain degree of clinical overlap [18]. In fact, MRH-related skin findings can mimic rheumatoid nodules or dermatomyositis-related Gottron papules. Histopathology of the nodular lesions, along with serology and radiology findings [10], can be very helpful in making the correct diagnosis and optimizing treatment [5].

Associated autoimmune diseases are reported to include: Sjogren’s syndrome, thyroid disease, systemic lupus erythematosus, multiple sclerosis [9], juvenile idiopathic arthritis, psoriasis, chronic focal granulomatous nephritis, myasthenia gravis, and immune thrombocytopenic purpura [5].

RA has been reported to be associated with MRH [4]. However, it has also been highlighted that MRH may be misdiagnosed as RA.

In the case we presented, the photo-distributed pattern, the facial erythema, and the concomitance with the summer season suggested a cutaneous photosensitivity, a condition already observed in association with MRH, because of the UV light-induced Koebner phenomenon [19]. However, it is also associated with connective tissue diseases. As reported above, in the literature, it is highlighted that approximately one-third of the patients diagnosed with MRH have confirmed concomitant autoimmune disorders, with positive autoimmune serologies for anti-Ro antibodies, anti-CCP, and more rarely, ANA [5]. Our patient exhibited clinical and laboratoristic manifestations consistent with an underlying autoimmune disorder, probably an early form of systemic lupus erythematosus [20]. In our case then, after a misdiagnosis with RA, we observed a clinical presentation compatible with dermatomyositis and, according to laboratoristic findings, with an early form of systemic lupus. The overlap with an autoimmune condition, from a clinical and laboratoristic point of view, was at first surprising. Therefore, we performed a literature search of PubMed using the key words “multicentric reticulohistiocytosis,” “case report,” and “autoimmune disease,” ranging from December 2021 to August 2022, producing eight previously reported cases, as summarized in Table 1.

MRH is characterized by skin and joint symptoms, which can present concomitantly or sequentially. It has been reported that the most frequent clinical features at presentation include cutaneous involvement with skin or mucosal papulonodular lesions, arthralgia, arthritis, unintentional weight loss, weakness/fatigue, dysphagia, and fever [5,9]. With skin and joint manifestations, it can mimic other rheumatic conditions, particularly RA and dermatomyositis [5]. The most frequently involved joints are reported to be the following: distal interphalangeal, proximal interphalangeal, and metacarpophalangeal joints, as well as wrists and knees. Shoulders, elbows, and ankles are also involved. MRH, when left untreated, can lead to a progressively deforming and destructive arthropathy, including contractures and arthritis mutilans.

Concerning cutaneous involvement, skin lesions may appear up to 3 years after the articular involvement: they usually consist of a slowing appearing acral, yellowish to brown-reddish papular-nodular eruption [10].

Systemic involvement has also been reported with MRH and includes pleural effusion, pericardial effusion, congestive heart failure, mesenteric lymphadenopathy, and urogenital lesions [5,10].

Of the 8 cases of MRH reported, 8 (100%) were characterized by articular manifestations, involving the MCP (1) and interphalangeal joints (5), knees (4), shoulders (3), elbows (2), wrists (3), and hips (1) (Table 2). Among these, 2 were diagnosed with RA. Articular manifestations were the first to be detected and were generally followed after many years by dermatological manifestations. The latter were harder to attribute to a unique condition, often requiring a histopathological exam to obtain the diagnosis. This was also true in our case, with a diagnostic delay of 11 years. Thus, joint manifestations appear to be particularly difficult to characterize, and the frequently reported association with seropositive RA may be due to misdiagnosis. Therefore, MRH may be considered, along with the differential diagnosis of rheumatic diseases with joint manifestation and positive anti-CCP antibodies (RA, juvenile RA, psoriatic arthritis, Sjogren’s syndrome) [28]. This fact is important to consider, as it may be linked to a less aggressive initial treatment than that required for MRH, leading to a rapidly progressive erosive arthritis. An interesting statistic taken from the cohort of 8 patients in the literature is the fact that autoimmune diseases seem to precede MRH diagnosis, or at least the diagnosis in its full form.

The cutaneous manifestations of the patients presented were compatible by appearance and distribution with the ones reported by other authors (Table 2). The hands were invariably involved, manifesting in the dorsum, or the hands and nailfolds of the fingers, and characterized by papular-nodular, firm, non-dolent lesions. In our case report, the papulose-nodular rash at first appeared to be compatible with other dermatological manifestation. Therefore, although extremely rare, MRH may be kept in mind when making differential diagnoses in regard to diseases such as dermatomyositis and lichen amyloidosis. Moreover, in the 8 cases reported in the literature, photosensitivity, facial erythema compatible with malar rash, and a photo-distributed pattern of the lesions were not mentioned, perhaps representing a peculiarity of our patient.

In the mini literature review presented, no signs and symptoms attributable to systemic involvement, other than autoimmune disease association, was found.

The autoimmune associated diseases mentioned in the literature were: Sjogren’s syndrome (4), primary biliary cirrhosis (1), vitiligo (1), thyroid autoimmune disease (1), seropositive RA (2), celiac disease (1), and autoimmune diabetes (1). Sjogren’s disease was the most frequently associated autoimmune disease, as previously reported in the literature. All the 8 cases presented affected women, who are known to develop autoimmune disorders much more frequently than men. This could be a possible bias to the overall interpretation of the prevalence of autoimmune disease in the cohorts of MRH patients, which are known to affect women more commonly than men.

It has been reported that MRH patients tend to manifest a wide variety of laboratory markers. These may include anemia, hyperlipidemia (30–50% of patients), and elevation of the erythrocyte sedimentation rate and C-reactive protein (50% of cases) [7,13]. Thrombocytosis and thrombocytopenia were also reported among the other findings. Several patients were reported to have positive autoimmune serologies for ANA (8 of 18 tested, 44%), rheumatoid factor (5 of 18 tested, 28%), anti-cyclic citrullinated peptide (8 of 18 tested, 44%), and anti-Ro antibody (3 of 4 tested, 75%) [5]. Moreover, positive anti-La, anti-dsDNA antibody, and perinuclear anti-neutrophil cytoplasmic antibodies (P-ANCA) were also reported. Low C-3 and CH50 were noted in 1 case [9].

In our case report, laboratory findings included high ANA titers (1:2560), as reported for cases 2 and 3 of our mini review, along with positive anti Ro/SSA antibodies, anti-CCP antibodies, and positive rheumatoid factor. Moreover, leucopenia and hypocomplementemia were also detected; however, they were interpreted as possible indicators of an early form of systemic lupus erythematosus, rather than as laboratoristic manifestations of MRH.

The early and accurate diagnosis of MRH is fundamental to prevent the progression of erosive arthritis to arthritis mutilans. Given the rarity of this disease, most of the information is contained in the case reports and, to date, there is no standardized treatment for MRH. Systemic glucocorticoids, NSAIDS, conventional disease modifying antirheumatic drugs (DMARDs) (including methotrexate and leflunomide), along with immunosuppressants, such as azathioprine, ciclosporin, chlorambucil, and topical tacrolimus, are reported to be the most frequently used treatments in MRH. Hydroxychloroquine, leflunomide, and anti-TNF biologics, such as etanercept, infliximab, and with adalimumab, have also been reported as therapeutic alternatives for the treatment of the disease [4]. Previously proposed treatment regimens have suggested the use of NSAIDS in the treatment of early and mild disease, or when treating the pediatric population. However, it has also been suggested that, given the generally aggressive nature of this disease, the early use of DMARD should be initiated once the diagnosis is made. Tariq et al. showed that the most effective initial DMARD is methotrexate, which seems to control arthritis symptoms in 28% cases and skin lesions in 38%. In cases with a contraindications to methotrexate, other DMARDs, such as leflunomide or azathioprine, may be considered. Hydroxychloroquine may be considered as a combined agent. Cyclophosphamide was found to be of significant benefit in 20% of cases, with complete arthritis resolution, and in 27% of cases of skin lesions. Additionally partial arthritis and skin disease control was seen in 40 and 45% of cases, respectively. Biologics may be kept in reserve for refractory diseases or in cases of major extra-articular manifestations. As MRH is associated with elevated TNF levels, it seems logical to use anti-TNF agents to antagonize the disease pathway. For this purpose, etanercept use was reported most often (6 cases), with clear disease control achieved in 3 cases and partial control in 2 cases. Adalimumab (2 cases) and infliximab (3 cases) were also shown to be effective. A recent case report mentions using anakinra (IL-1 inhibition), which allowed for control of the disease [29]. Bisphosphonates may play a role in erosive arthritis, and these could be considered as add-on agents in cases of poor disease control or in cases of concomitant osteopenia (or osteoporosis) and/or steroid use [9].

In the literature, the use of JAK inhibitors, such as tofacitinib and upadacitinib, is also reported. Tofacitinib was reported by Bruscas Izu C. et al. to be a possible therapeutical alternative when antirheumatic drugs classically used in MRH, namely prednisone, methotrexate, and etanercept, fail to improve articular and skin manifestations [30]. Moreover, a severe case of MRH, resistant to first-line therapies, showed a marked improvement in cutaneous lesions and arthritis following treatment with upadacitinib, as reported by Niaki OZ et al. [31]. Although an empiric algorithm for the treatment of MRH exists, a large, single-center case series is needed to clarify its actual incidence, associated systemic disorders, and therapy responses.

Conventional treatments for MRH are represented by immunomodulatory treatments which may simultaneously treat associated MRH and autoimmune diseases, most of the time with success. In the mini review under consideration, the patients received: NSAIDS (3), corticosteroids (6), d-penicillamine (1), cyclophosphamide (2), chlorambucil (1), methotrexate (4), hydroxychloroquine (3), and bisphosphonates (3). All the patients received benefit from the first or second line therapies administered. Our case report, including a diagnostic delay, was characterized by a complicated therapeutic process, the patient being intolerant/resistant to first line treatments. NSAIDs, prednisone, methotrexate, adalimumab, and baricitinib were followed by the administration of an anti-IL6 biologic (sarilumab, tocilizumab), with improvement in both joints and dermatological manifestations. The therapeutic approach pursued was found to be in line to that of the algorithm proposed by Tariq et al. in their review [9]. Although these drugs have different mechanisms of action, they all exhibit an immunomodulatory effect. While MTX increases the sensitivity of T cells to apoptosis [32], adalimumab, baricitinib, sarilumab, and tocilizumab target specific cytokines, reducing the inflammatory response. In particular, adalimumab combats the destruction process of the joints observed in RA by reducing the effect of overproduced TNF [33], while tocilizumab competitively inhibits the binding of IL-6 to its receptor, inhibiting the cascades involving the Janus kinase-signal transducer, modulating joint inflammation [34] It has been reported that interleukin 6 is overexpressed in the rheumatoid synovium and in the giant multinucleated cells of the MRH inflammatory infiltrate. Therefore, sarilumab and tocilizumab have been proposed as specific immunomodulators to treat MRH refractory patients. However, there seems to be a two-way relationship between TNF and IL-6, as it has been reported that a blockade of IL-6 successfully followed anti-TNF therapy, in some of its indications [35].

## 4. Conclusions

This mini literature review highlights the frequent delay in the diagnosis of MRH, mainly due to misinterpretation of joint manifestations, which usually precede dermatologic symptoms by many years. A high vigilance for an association with autoimmune diseases is also needed. In the case report presented, it was particularly difficult to distinguish which manifestation were attributable to MRH and which to an autoimmune condition. Although the etiology and physiopathology of MRH are unknown, the association with autoimmune diseases reported in the literature and observed in our case may suggest a pathogenetic relationship. This may also be supported by evidence reported concerning the efficacy of anti-inflammatory/immune-modulatory treatments. Therefore, an immunological disorder could underly and anticipate MRH.

## Figures and Tables

**Figure 1 jcm-11-06529-f001:**
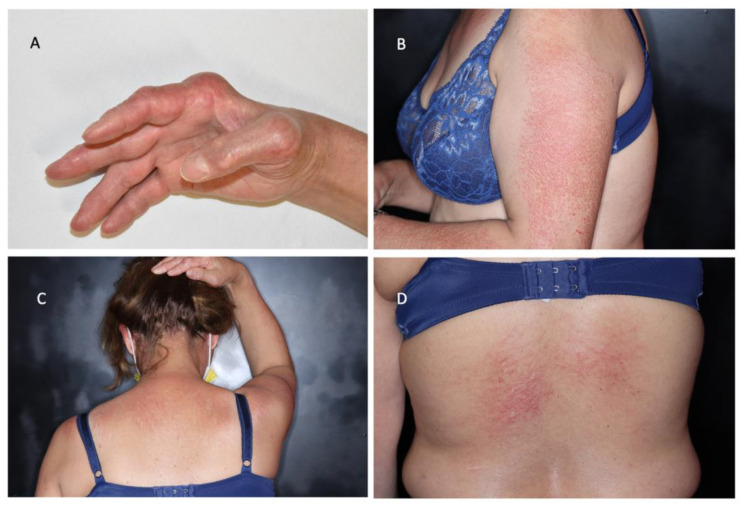
Clinical photos showing: (**A**) hand deformities; (**B**–**D**) photo-distributed erythematous papular rash of the upper body, with signs of scratching.

**Figure 2 jcm-11-06529-f002:**
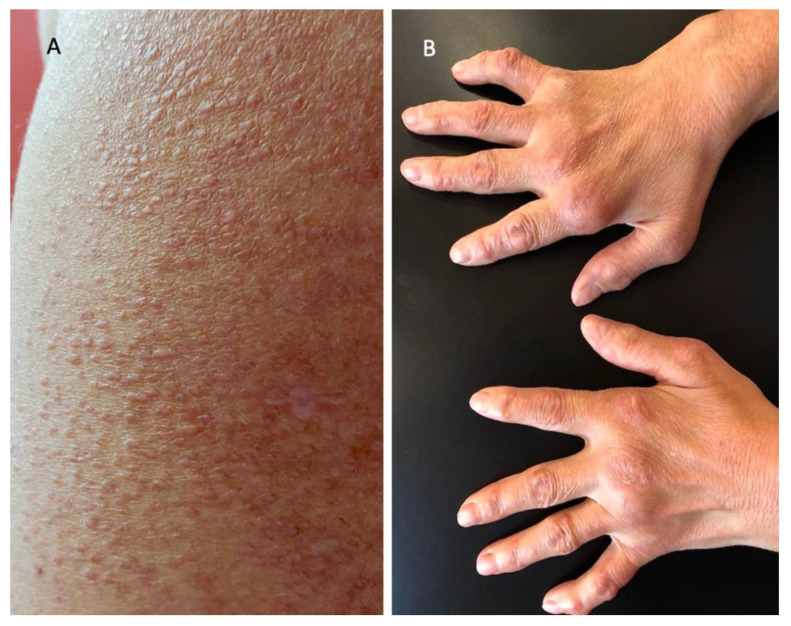
Detail of the lesions of the skin: (**A**) non-confluent papule of the upper body; (**B**) periungual and periarticular skin-colored nodules.

**Figure 3 jcm-11-06529-f003:**
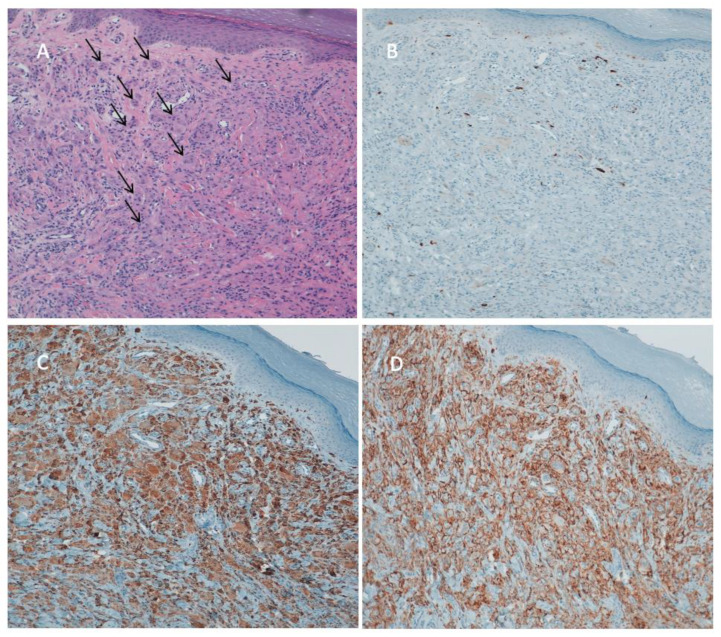
Immunohistological pictures: (**A**) histiocytes and multinucleated giant cells in the dermis indicated by the arrows; (**B**) neurone-specific enolase and Sangtec 100 (S-100) negative staining for infiltrating cells; (**C**) cluster of differentiation 68 (CD68) and (**D**) cluster of differentiation 45 (CD45) positive cells.

**Figure 4 jcm-11-06529-f004:**
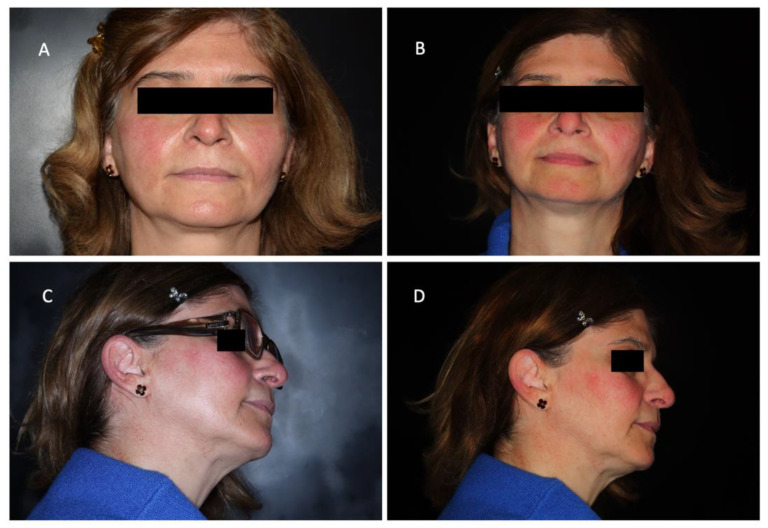
Clinical photos showing the erythema of the face, with malar rash-like distribution (**A**) and sparing of the nasolabial fold (**B**); the papular rash involving the ears (**C**) and sparing the cheeks (**D**).

**Figure 5 jcm-11-06529-f005:**
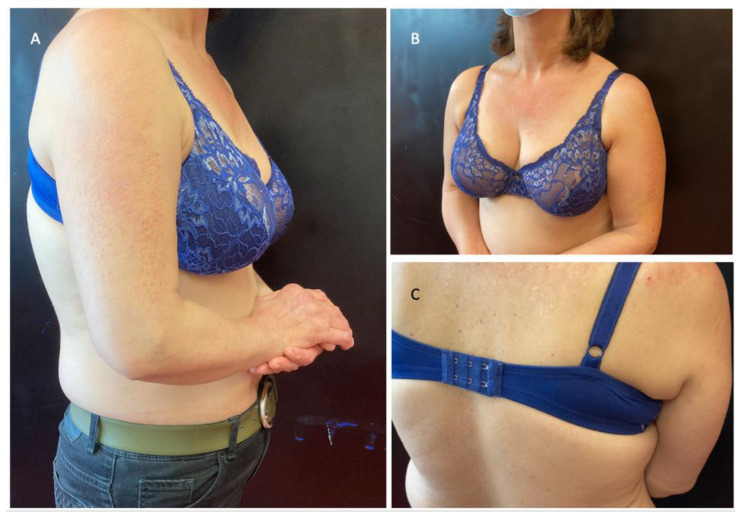
The patient after treatment with tocilizumab: the papules drastically reduced in number (**A**) and the erythema was less noticeable (**B**,**C**).

**Figure 6 jcm-11-06529-f006:**
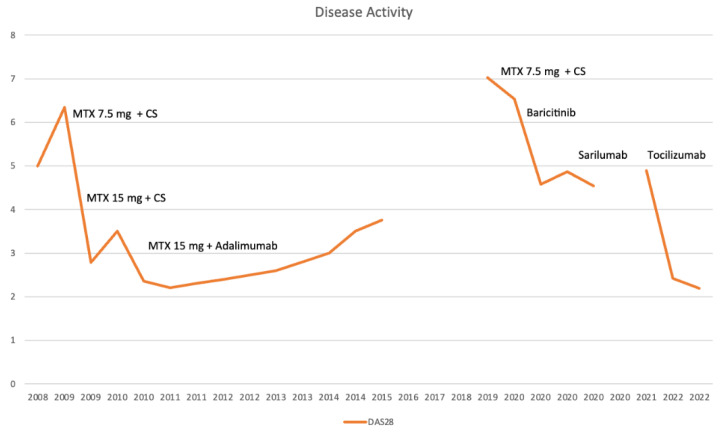
Modified Disease Activity Score (DAS28) during follow-up. Therapy and disease activity evaluation had to be interrupted between 2015 and 2019, as the patient moved to another country. NSAIDs = non-steroidal anti-inflammatory drug; CS = corticosteroids; MTX = methotrexate.

**Table 1 jcm-11-06529-t001:** Cases of MRH (Multicentric reticulohistiocytosis) associated with autoimmune diseases reported in the literature up until August 2022. Data adapted from the following articles: (1) MRH associated with primary biliary cirrhosis: successful treatment with cytotoxic agents [21]; (2) MRH associated with subclinical Sjögren’s syndrome [22]; (3) MRH in a Taiwanese woman with Sjögren syndrome [23]; (4) MRH associated with RA (rheumatoid arthritis) [24]; (5) Paraneoplastic multicentric reticulohistiocytosis associated with a celiac disease [25]; (6) MRH: a systemic osteoclastic disease? [26]; (7) Case report of an association with erosive RA and systemic Sjogren syndrome [18]; (8) A Rare Case of MRH with Concurrent RA [27]; AIDs = autoimmune diseases; y.o. = years old; PBC = primary biliary cirrhosis; AI = autoimmune; RF = rheumatoid factor; CS = corticosteroids, MTX = methotrexate; TG = transglutaminases; DM = diabetes mellitus.

	Timing of Clinical Manifestations	AIDs Associated	Serology	Timing of Diagnosis	Therapy
Case 1. Caucasian woman, 56 y.o.	Joint manifestations preceded dermatological manifestations	Vitiligo, PBC, AI thyroiditis	Positive thyroid microsomal and thyroglobulin antibodies	MRH diagnosis following AIDs diagnosis	NSAIDs, D-penicillamine, cyclophosphamide, chlorambucil
Case 2 Caucasian woman, 60 y.o.	Joint manifestations preceded dermatological manifestations (7 years)	Sjogren syndrome	Positive ANA (1:80) and anti-Ro/SSA antibodies	Pre-existent Sjogren’s syndrome	Cyclophosphamide, CS, sodium aurothiomalate
Case 3 Taiwanese woman, 59 y.o.	Joint manifestations preceded dermatological manifestations (2 years)	Sjogren’s syndrome	Positive ANA (1:640), anti-Ro/SSA and anti-La/SSB	Pre-existent Sjogren’s syndrome	CS, hydroxychloroquine
Case 4 Japanese woman, 67 y.o.	Joint manifestations preceded dermatological manifestations (20 years)	Seropositive RA	Positive RF	MRH diagnosis followed RA diagnosis (22 years later)	MTX, CS
Case 5 Caucasian woman, 68 y.o.	-	Celiac disease	Positive anti-endomysium, anti-TG, and anti-gliadin		CS, MTX
Case 6 Caucasian woman, 40 y.o.	-	Sjogren’s syndrome, RA	Positive RF and anti–CCP	Sjogren’s diagnoses preceded MRH diagnoses	CS, hydroxychloroquine, MTX, intravenous infusions of zoledronic acid
Case 7 Caucasian woman, 50 y.o.	Concomitant	RA, Sjogren’s syndrome	Positive RF, ACPA, and anti-Ro/SSA	MRH complicated by Sjogren’s syndrome	MTX, CS, alendronate
Case 8 Caucasian woman, 60 y.o.	-	Type 1 DM	Anemia, elevated VES, PCR, RF, anti- CCP titer	-	NSAIDs, MTX, CS, hydroxychloroquine, ibandronate

**Table 2 jcm-11-06529-t002:** Joints and cutaneous manifestations in 8 cases of MRH reported to be associated with autoimmune diseases in the literature. y = years; AI = autoimmune diseases; MCP = metacarpophalangeal; PIP = proximal interphalangeal; DIP = distal interphalangeal.

Cases of MRH Associated with AI Disease	Articular Manifestations	Dermatological Manifestation
Case 1	A 4 y history of pain and stiffness affecting the knees, wrists, MCP, PIP, and DIP joints	Asymptomatic verrucous nailfold swellings on several fingers
Case 2	A 7 y history of polyarthritis and stiffness involving the knees, wrists, fingers, shoulders, and elbows, with diagnosis of RA	Multiple small, reddish-brown cutaneous nodules on the fingers
Case 3	A 2 y history of polyarthropathy affecting mainly the knees and the small joints of the hands	Itchy papulonodular rash affecting the dorsa of the hands, the neck and the helices of the ears
Case 4	A 20 y history of pain, swelling, and tenderness in the elbows, knees, shoulders, and hip joints, with a diagnosis of RA with positive rheumatoid factor	Papules on the face, especially on the ears and the dorsum of the hands and fingers
Case 5	Inflammatory arthralgias, without signs of arthritis, involving the fingers, wrists, and shoulders	Erythematosus/violaceus papulonodular lesions of the face, décolleté, hands, and knees
Case 6	An 8 m history of joint symptoms	Non tender skin-colored papules and nodules over the PIP and the DIP joints, elbows, forehead, and ears; characteristic “coral beads” over the proximal nailfolds
Case 7	Previous diagnosis of RA complicated by an extension of damage to the DIP joints	Rosy purple, smooth, firm, nodules, initially periungual, then spreading throughout the body and the dorsal surface of the fingers, elbows, pavilions, ears, scalp, and neck
Case 8	A 30 y history of symmetrical polyarthritis, with morning stiffness	Papulonodular pruritic skin lesions over the scalp, trunk, extensor aspect of the elbow, dorsum of the hands, and pinnae

## Data Availability

Not applicable.

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
