# Peer review of "Multicentric Reticulohistiocytosis Associated with an Early Form of Systemic Lupus Erythematosus: A Case Report of a Rare Disease, with Mini Review of the Literature"

_jcm, 2022, doi:10.3390/jcm11216529_

Round 1

Reviewer 1 Report

The case report of MRH from Mariotti et al reported a successful treatment of a 50-year-old woman with MRH. The report comes with a comprehensive discussion, which is highly appreciated. However, the manuscript contains many language and grammar issues. While the treatment of tocilizumab greatly ameliorated the skin manifestations, it would be helpful if doses and durations of all treatments (including previous non-responsive treatments) are provided. It will be more valuable if underline mechanisms of different treatment options are discussed. 

Minor points:

1. It would be helpful if giant cells are pointed in Figure 3A.

2. It's contradict to say MRH interests every internal organ (in abstract), but it's actually rarely affect internal organs (Abstract, pg 2).

3. Pg 1, introduction, "RH are a group" should be "RH is a group". Suggest to check English all through.

4. Pg 9, line 30, An interesting data from our cohort of 8 patients. This is a case report. Where is your cohort?

5. Discussion, first paragraph. "Authors should discuss the results ... may also be highlighted". Where is this coming from? manuscript requirement? or accident incorporation. 

Reviewer 2 Report

This is an interesting case presentation, followed by brief literature review, describing autoimmune manifestations that precede multicentric reticulohistiocytosis. There is a particular attention to the pattern that articular manifestations preceded dermatological ones, which was consistent with the other cases.

Major

1. Could the authors elaborate better the history of treatment? Especially with regards to the RA diagnosis in ?2009 followed by unsuccessful treatment. How about the lab values, when were they captured during the clinical course?

2. Could the authors provide disease activity scoring that shows disease progression? It might be a good idea to visualize it in a simple chart/plot. With chronic joint manifestations as shown in the figures, it seems possible that disease activity has been documented in this patient's medical record.

Minor

1. Please remove the first three sentences under Discussion which were included by mistake.

2. Please make the nomenclature consistent with regards to Sjögren's syndrome.

3. Please clarify the RA history with regards to rheumatoid nodules and RA diagnosis (years as currently written: 2018 and 2009).

4. Please appropriately address patient's privacy with regards to Fig. 4 (e.g black out the area surrounding eyes).

Round 2

Reviewer 2 Report

Everything was clarified, thank you very much.